# Really, Truly Trans and the (Minor) Literary Discontents of Authenticity

## Aaron Hammes

English Department, City College of New York, New York, NY 10031, USA; ahammes@gradcenter.cuny.edu

**Abstract:** Identity formation, questions of identity, shifting identities, perceived deviant identities, and reactions (social, political, cultural, individual) to them are the stuff of Bildungsroman as well as more "experimental" subgenres of long-form fiction. For minority/minoritized subjects and authors, questions of identity take on a different pallor: their work is expected to engage with questions of identity according to either or both how their subject position confronts marginalization and otherness, and how their subject position conditions every experience they have in the world, both inside and outside community. This inquiry investigates how contemporary transgender minor literature constructs dis/identity through authenticity. Imogen Binnie speculates in her 2013 novel *Nevada* on the concept of "Really, Truly Trans", a cipher for identity policing and presumptions of sex–gender authenticity, based on cisnormative characteristics and, occasionally, inter-community phobias and proscriptions. More recently, Torrey Peters challenges measures of trans authenticity through both her titular detransitioner and his former partner in *Detransition, Baby*. Trans minor literature is an ideal testing ground for phobic public presumptions around "authentic" sex–gender and anti-identitarian strategies of those who are forced to confront purity tests and exclusion or suppression on grounds of authenticity, and each novel presses at phobic majoritarian dictates of authenticity and its presupposed value.

**Keywords:** minor literature; transgender novel; detransition; sex–gender; identity

## 1. Damage and Really, Truly Trans

Maria Griffiths, anti/heroine of Imogen Binnie's (2013) recently reissued *Nevada*, has a terrible job in retail in New York City, a partner who knows Maria fakes her orgasms and in turn pretends to cheat on Maria, just to see how she (or if) she reacts, and friends who expect to hear from Maria only when times are particularly dire. The job represents the locale in which a particular valence of Maria's passing—she notes that her "retail persona is impressively unbitter" (p. 18)—is mostly successful, though, as in most other areas of her life, she cannot pass as someone who cares. She reflects on the "shitty job" as she plans to leave it: "I'm the sort of person who has too much self-regard to stay at this job, too, except I guess I'm all damaged. Meaning: trans" (p. 41). "Damaged" carries a multiplicity of meanings, which are explored throughout the novel, from Maria's perspective, that of (ex)girlfriend Steph, trans friends Kieran and Piranha, and, later, sex–gender-agnostic/questioner James. Here, Maria's "damage" is set in contradistinction to her self-regard (which would demand quitting the job) and is defined out of her trans identification and subject position. But she goes on to consider what variety of trans constitutes her damage—why she is stuck in this particular rut—and it may come down to being insufficiently, incorrectly, or even inauthentically trans, according to the phobic public institutions which level such judgments. She notes: "There is this dumb thing where trans women feel like we all have to prove that we're totally trans as fuck and there's no doubt in our minds that we're Really, Truly Trans. It comes from the fact that you have to prove that you're trans to psychologists and doctors: the burden is entirely on your own shoulders to prove that you're Really Trans in order to get any treatment at all" (ibid.). The damage is in

Maria's trans-ness, as well as the attendant tests of authenticity and commitment. Other versions of these metrics of authenticity and concomitant "damage" are shot through trans minor literature—minor in part due to its thematizing the minoritarian and refusing to cotton to majoritarian presumptions regarding authentic characters and how to measure them. Deleuze and Guattari (1975) instantiated the category of minor literature to consider the works of Franz Kafka and his position within Central European and Germanic literature. Their three characteristics of this literature include its ability to deterritorialize language, its sense of political ubiquity for its minoritarian subjects and authors, and its collective value for these communities and individuals. Hammes (2022) has written elsewhere about the ways in which minor literature can expand on potentials for minoritarian identity and expression, but this essay will further test the unique purchase trans minor literature can have on minoritarian experience, stressing and complicating presumptions around identity formation and expression. Majoritarian dictates of authenticity are, this essay holds, another ripe ground for deterritorialization and remapping. As such, this inquiry considers what (if anything) it might mean to be authentic to sex–gender: is it equivalent to "passing as", or simply being "true to oneself"? Can authenticity be measured as the level of commitment, and if so, commitment to what: what one seeks to become, the becoming itself, or neither? Finally, does Really, Truly Trans promote the kind of damage Maria cites (though she will leave the shitty job, and the city, shortly after this moment in the plot), or can it be a bulwark against it? I will build up a few theoretical options for defining authenticity, while considering how defying, transgressing, embracing, or abdicating the sex–gender systems of long-form fiction in trans minor literature operates, primarily through *Nevada* and Peters' (2021) *Detransition, Baby*.

## 2. Authenticity as Fidelity and Process

Jacques (2019) notes Maria Griffiths' "internalized transphobia" (p. 269) as the latter avers that she "fucking hate[s] everybody else who's trans, and [she doesn't] want to deal with it." Binnie's novel is an intentionally incomplete, mutated hero's quest or *Bildunsgroman*, following Maria from a broken relationship with Steph in New York to desolate small-town Nevada and (potentially trans) James, whom she meets completely by chance in his own shitty job, at a megastore in Star City. *Detransition, Baby*, on the other hand, is a contemporary nonnuclear family novel, tracing the aftermath of trans woman Reese and detransitioned man Ames's relationship, now including the possibility of raising the prospective child with which he has impregnated his cis boss, Katrina. Reese has her own "internalized transphobic" thoughts, albeit in a different register to Maria's above: "For years, Reese had a rule: Don't date other trans women. It was a hypocritical rule. Had anyone else ever ruled her as ineligible for dating on account of her gender, she'd have cried transphobia. But in her own secret heart, the idea of dating another trans woman repelled her" (p. 61). These literary moments speak to a variety of authenticity that resists de facto community or relationships with others who share minoritarian subject positions. This section traces a few influential concepts of authenticity as fidelity to one's own precepts—even those which appear hypocritical—and how trans minor literature ascribes to or challenges presumptions of the Really, Truly authentic.

Guignon (2010) defines the process of the contemporary "ideal of authenticity" as follows: first, get in touch with any "real self we have within" (p. 6), a task requiring "introspection, self-reflection or meditation". Second, one must express their "unique constellation of inner traits" outwardly, "to actually *be* what we *are*" in relationships, careers, and practical activities. Maria's, and perhaps Reese's, "damage", is certainly not through any lack of introspection; each spends the bulk of her narrative considering her own subject position and how she does and does not relate to being trans. Still, this ideal requires the existence of a "real self" to pursue and seems to imply that one is not "naturally" in touch with it—it must be chased. The back half of *Nevada*, for instance, is a road novel, Maria jetting across the country to the titular state with minimal funds, a "borrowed" sedan, and a glovebox full of narcotics. Is she running to or from, or neither?

Further considering the pursuit metaphor, Sartre (1993) contraposed authenticity with guilt and bad faith, the latter of which "seeks by means of 'not-being-what-one-is' to escape from the in-itself which I am not in the mode of being what one is not. It denies itself as bad faith and aims at the in-itself which I am not in the mode of 'not-being-what-one-is-not.'" (p. 70). The animus of bad faith is to seek, to escape by not being oneself, but neither Maria nor Reese acts in this kind of bad faith—they can neither help being what they are, nor seek to escape from it. Even Ames, the titular detransitioner, is seen to be in bad faith only by former lover Reese herself, who cannot believe he would want to back away from woman-ness. Still, Sartre goes on to note it is in "self-recovery of being which was previously corrupted [ . . . ] we shall call authenticity" (ibid.), though it is not "through unauthenticity that human reality loses itself in the world" (p. 200). But bad faith and "losing reality" can be self-preserving as well; while still in Brooklyn, Maria finds herself in a diner, about which the narrator notes that in her "trite lifelong quest for authenticity, Kellogg's still kind of rates a blip" (p. 48). The place is authentic, perhaps, but Maria's experience is unusual based on her (and the other patrons') subject position. She thinks, tautologically, "These are the situations where, if you are trans, you are going to get read as trans, and it is going to be a situation" (p. 49). But there is no situation this time, which causes Maria to speculate on the extent to which her pursuit (or her damage) has foreclosed the likelihood of confrontation. She continues "Nobody notices her. It's funny. Nobody ever does any more. It's just that when they used to, they were so vocal about it that still, to this day, you worry" (ibid.). It is not out of bad faith that Maria "can't help but wonder what people see when they look at you" (ibid.), but rather exhaustion at the fact that "if people look at you and figure out that you are trans, they are pretty eager to tell you" (ibid.). Whether or not these people are registering the authenticity of Maria's trans identity, they are taking license to inform her about who she is and what that signifies. Really, Truly Trans seems to include the ability of cis people to read a trans subject sufficiently to engage them in conversation about their sex–gender identity, and "transition story." Maria wonders, "can I get twenty minutes where I don't think about being trans, please?" which is followed by the realization that "she is literally halfway toward twenty minutes where she doesn't have to think about being trans" (p. 50). She has, of course, been thinking of that the entire time sitting in the authentic diner, but she isn't forced to by anyone else.

Other accounts of authenticity as fidelity/pursuit are less dependent on outside endorsement. Bauer (2017) traces a "volitional account of authenticity" (p. 569), which stems from a conception of authenticity as an expression of autonomy and a desire to be "a self" and being true to "who you want to be" (p. 572). This locus of control is then linked to Charles Taylor's "responsibilization of authenticity" and one's behavior's "appropriateness to moral law" (p. 573). To this, Bauer compares an "epistemic conception of authenticity," which defines authenticity as "truthfulness toward oneself and about oneself in word and deed" (p. 574). Bauer concludes "that self-knowledge is not only important for being authentic, but also for being prudent, for acting efficiently, for giving reasons for one's actions or for being wise in a rather abstract philosophical sense" (p. 576). Maria and Reese challenge each of these definitions in some measure, casting doubt on the value of these sorts of authenticity, their need to pursue them, or both. Maria expresses her autonomy predominately through a sloughing off of responsibility, reinforcing her affected lonerism until taking on a kind of tutelage relationship with James. At this meeting, *Nevada* shifts to a face-to-face exercise in recognition and disidentification—Maria feels some kind of responsibility to both present an appropriate persona of trans woman-ness to James, and to coax him into more seriously considering his own sex–gender identity. Prior to meeting Maria, James's sense of gender has been locked up in sexuality: a frustrated relationship with a feminist girlfriend and surreptitiously buying dresses and researching sissy fetishes online. Even to James, Maria obviously does not fit stereotypes of autogynephilia or confused homosexuality; he is assured that she is (Truly) Trans because he asks. What exactly he thinks this signifies evolves throughout the last chapters of the novel and is left at best incomplete in the end.

Nonetheless, who Maria, or Reese, wants to be is a concern unmoored from valuative certainty: Maria has survived transition, sexual disinterest and frustration, and somewhat self-enforced poverty without any specific aspiration (that the reader knows of). Reese has been solicited as a mother (to Amy and other trans women), sister (to her household peers), secret mistress/sugar baby (to Stanley), and mother again (to Ames and Katrina's child). Authenticity as fidelity to whom Reese wants to be would require Reese (or the reader) to know who that is. It is, instead, a moving target.

Varga (2014) charts a more continually dynamic version of the quest for authenticity, closer to living artfully than working towards an ideal of self. Under this conception, "authentic selfhood is achieved, not by introspection, but by continuously reworking and recasting the elements of one's life into an aesthetic whole" (p. 53). Varga quickly dispenses with this vision of the creation of the self as necessarily evaluative, "involving ideas about why such a life is better, 'higher', or worthier than others" (p. 54). But this all depends on the model—the aesthetic genre, perhaps. The reworking and recasting of Maria's life may be cyclical (Steph charts the repetitiveness of Maria's annual escape (p. 118)), but it proceeds from the subject position of a trans woman. While Steph notes that "nothing changes" after Maria's adventures, Maria herself reflects on the ways people address her differently, and she, in turn, is more confident in how she comports herself towards them— at least the people with whom she does not share any intimate, lengthier bond. In any case, rather than "responsibilize" authenticity, Maria decides at a certain point that "she needs to be extremely irresponsible in her life from now on" (p. 95). Reese, in a slightly different valence, compares having unprotected sex with a seroconverted HIV-positive man to the risk of pregnancy ("an analogue to a cis woman's life changer" (p. 7))—irresponsibility to her own continued persistence feels like an analogue to authenticity. These rejections (or even inversions) of responsibility are authentic in their fidelity to infidelity: why be responsible to majoritarian dictates of the "good life" or survival when these so routinely elide or erase minoritarian subjects and their experiences?

### 3. Authenticity as Black Sheep

Early American hardcore act Minor Threat reflected a particular stream of punk authenticity and commitment. For one thing, band members founded Dischord Records, which produced and released music strictly by DC-based acts at discount prices and without major label distribution. For another, the band espoused a "straight edge" ethos, abstaining from drugs and alcohol and noting ways in which these commitments rendered them outsiders, even within nascent hardcore scenes they helped foster. The brisk title track of their lone studio LP, "Out of Step" (Minor Threat 1983), reinforced both the straight edge mentality ("I don't smoke/Don't drink/Don't fuck/At least I can fucking think") as well as how this sets them against the zeitgeist ("I can't keep up [ . . . ] out of step with the world"). Guignon writes about a version of "being authentic" which "means being fundamentally and unavoidably out of step with the mainstream" (p. 76), something Maria and Reese embrace in different ways, though each filtered back through their minoritarian subject positions. A few years after the breakup of Minor Threat, vocalist Ian MacKaye formed Fugazi, featured on one of the two CDs Maria has in Steph's "borrowed" car as she prepares to escape west. Maria "turns it up. She's sixteen, but she's the right sex this time, and it feels mostly liberating and exciting but also a little sad" (p. 125). This sadness turns out to be "empathy" for sixteen-year-old Maria, whom current Maria wants to tell: "your life is going to get better than you can imagine right now" (ibid.). This is not a moment of Maria signifying her subcultural credentials to anyone else, more a matter of reflecting on her own "progress"—out of step as it may be—toward a better life.

Bettcher (2017) details "the basic denial of authenticity", a "form of transphobia" premised on "identity enforcement" (p. 235). Maria relates moments of microaggression and (justified) fears of transphobic interactions, but her identity enforcement is largely internalized, often considering her position within her own community. Reese, on the other hand, has had a great deal asked of her in her identity as a trans woman—a different

kind of identity enforcement which results in frustrations around being hidden in her sexual relationships with cis men, alienating Amy (prior to detransitioning as Ames) at the end of their relationship, and having her invitation as co-parent revoked. What Bettcher has in mind is identity enforcement that takes "for granted singular, fixed meaning of gender terms", noting that such accounts "cannot plausibly provide a liberatory theory" (p. 247). While Maria feels a certain measure of liberation in her escape, and Reese in her potentially (and actually) destructive relationships with cis men, Bettcher's disavowal of accounts that "aim to justify the categorization of trans people by appealing to the dominant meanings" which "implies acceptance of a marginalizing asymmetry between trans and non-trans people from the beginning" (ibid.) is experienced by both women. Furthermore, both Maria and Reese develop different strategies to redress this asymmetry, Reese in her amative relationships, and Maria in her escape from them. With nearly a decade more trans-information-sharing and community building, though, Reese puts a finer point on the self-knowledge inherent in her trans identity: "How is it, Reese wonders, that a bunch of New York men wearing flannel and slamming whiskey in a cabin is seen as a sorely needed release of their barely tamed and authentic manliness, but when she, a trans, delights in dolling up, she's trying too hard?" (p. 274). Cis men acting according to majoritarian dictates of "masculine" behavior expresses their "authentic" sex–gender identity, whereas any attempt by Reese to espouse analogously "feminine" behavior is read as inauthentic. But this concept of authenticity is not simply cloth deep: "It's not that Reese thinks her desire to dress up reflects some authentic self. It's just that, unlike bros, she's willing to call dress-up time what it is" (ibid.). But what is at stake for the bros calling their pronounced masculine behavior "dress-up" or not? Does their unwillingness to do so imply that their authenticity as "men" stands up only insofar as it meets the social codes that they help authorize? Conversely, does Reese's willingness decouple sex–gender from outward presentation, or does it imply that authenticity is the willingness to name roles for what they are? Really, Truly Trans could as soon threaten in its exposure of the contingency of outward sex–gender markers as challenge what those markers actually denote. Reese has considered this state of play before: "She decides for the ten thousandth time that heterosexual cis people, while willfully ignoring it, have staked their whole sexuality on a bet that each other's genders are real" (p. 275). These cishet people emblematize Gayle Rubin's (1984) sex/gender/sexuality system, "staking" sexuality on gender, rather than the other way around, and are not simply ignorant of it, but willful in their failure to recognize it. Where does authenticity stand in relation to majoritarian publics ignoring the fictions that undergird their sexuality and desire? Rubin saw gender as a "social product;" Reese reads (cis)sexuality as the byproduct of a fictitious system. Furthermore, it is people in her subject position that could show them: "If only cis heterosexuals would realize that, like trans women, the activity in which they are indulging is a big self-pleasuring lie that has little to do with their actual personhood, they'd be free to indulge in a whole new flexible suite of hot ways to lie to each other" (ibid.). Insincerity, inauthenticity, and lying could lead to the indulgence of untold desires.

Maria, on the other hand, seeks to escape the desires of others: her partner's desire for homonormative queer intimacy, her friends' desires for reciprocal care, lower Manhattan, and Williamsburg's desires for carefully curated queerness and counterculture. On this latter point, Harry S. Rubin (1998) notes ways in which "In order to find a place for themselves at the queer table", trans people "have sometime figured themselves or been refigured by others to accommodate the privileged terms of queer rhetoric" (p. 275). This rhetoric has, in Rubin's view, a relatively limited vocabulary for trans people, which suggests "To be a queer transgender man or woman often means being self-reflexive about gender as a construct, purposely gender fucking to disrupt naturalized gender", or, put another way, "living with illegible bodies" (ibid.). Illegibility-as-authenticity is in Maria's past, as she notes that "Androgynous fag" was "a look she tried for when she first started transitioning" because it "doesn't disrupt strangers' worldviews much and theoretically they will just ignore you" (p. 49). Rubin might account for this as

"speaking the unnaturalness of legible reinscription" (p. 275), as Maria's androgynous liminal state is limited. Furthermore, in whatever other ways she might feel compromised, her commitments to her own trans woman-ness are sound, even as she explores what that means.

The flexibility of sex–gender presentation and identity are central to the ways in which authenticity can be "out of step" with majoritarian convention. While Minor Threat defined themselves as outside of society due to their abstinent relationship with drugs, alcohol, and sex, genderqueer hardcore group G.L.O.S.S. asserted both subject position and disdain for its presumptions in their very name, Girls Living Outside Society's Shit. Vocalist Sadie Switchblade (2015) was direct about the band's intentions ("We're a hardcore band interested in inciting violence, and we wanted to have a name that emphasized our unwillingness to acquiesce to social expectations."), and the group broke up shortly after being offered a five-figure record deal with a prominent punk label. Their earliest lyrics (G.L.O.S.S. 2015) speak most directly to the authenticity of resisting illegibility. Switchblade sings, "expected to be grateful/trapped in the lens of the cis-gaze" which is paid down by the chorus "Masculinity was the artifice, rip it away/Femininity, always the heart of us/Trans girls be free." Bettcher notes that fixity precludes liberation, which for this investigation begs the question of whether dictates of authenticity are liberating or imprisoning. While G.L.O.S.S. casts off artifices of masculinity, Maria ponders the ways in which her severing ties with Steph and her New York life may be "some straight dude bullshit, the self-sufficient loner" (p. 37). But this is simply adopting a calcified representation; Bornstein (1994) notes, "When I get too tired of not having an identity, I take one on: it doesn't really matter what identity I take on, as long as it's recognizable" (p. 39). This approach could be deemed the pragmatics of legibility, with Bornstein at intervals tiring of being without an identity, even as she understands its impermanence. Maria's determination of "straight dude bullshit" thinking does not end up preventing her escape, and even in this moment "She felt liberated for a second" (p. 37). Harry S. Rubin, too, recognizes the potentiality for deterritorialization and rewriting as he asserts "We transsexuals are agents who actively become the selves that we have always been to ourselves by refusing and resignifying the categories given to us by discourse" (p. 278). Not, mind, the selves "we seek to be," or "could become," but "have always been," and if that is somehow out of step with phobic public expectations, then perhaps authenticity is resistant. Guignon expresses something similar regarding out-of-step authenticity: "to be in touch with something that is concealed to the people who accept the outlook of society;" that is, "to be authentic is already to be asocial" (p. 76).

Truly Trans is authenticated in at least a few different registers, depending upon who is compelled to authenticate (or not). Jacques writes about this sort of authentication across her work, from experiences with NHS-supported gender-affirming surgeries in *Trans: A Memoir* (Jacques 2016) to the polygeneric *Variations* (Jacques 2021). Bauer notes, "Beyond her self-authentication as an authentic personality, an authentic person also depends on being authenticated by others. The mutual authentication of persons as persons presupposes a realm of trustfulness as the individual self-authentication presupposes truthfulness about and toward oneself" (p. 577). In Jacques, a physician's authentication is an almost comically reductive form of identitarian gatekeeping. *Variations'* "Standards of Care" centers on trans woman Sandy and her experiences with gender-affirming surgery in late-1970s London. Truly Trans here sits at the intersection of medico-statutory "expertise" and cultural norms which go beyond any binary to a specific archetype of not just woman-ness, but trans woman-ness. Sandy transcribes in her diary her physician's judgment that her "dress and comportment" are "not yet satisfactory for someone living as female" (p. 145). The doctor's further comments read like a conduct manual: Sandy should be "more discreet" (p. 146) about being trans, wear dresses rather than trousers, lose weight and quit smoking, and take any harassment in stride (Sandy can "hear [Dr.] Randell saying *it's all part of the Test*" and thinks "perhaps it is" (ibid.)). As such, she ends up caught in a bind not dissimilar to Reese's note about masculine cis men and not-feminine-enough/too-feminine trans

women: "I'd give them what they want but it's so hard to tell. Do they want me to be 'more discreet' or are they asking me to get into a beauty contest with [Irish drag performer] Danny sodding La Rue?" (ibid.).

Stone's (1992) "The Empire Strikes Back" puts pressure on these sorts of external dictates of authenticity. Stone notes that "Given this circumstance in which a minority discourse comes to ground in the physical, a counterdiscourse is critical" (p. 295). In this instance, the minority discourse is around trans woman-ness and the mandates of authenticity. Stone goes on: "But it is difficult to generate a counterdiscourse if one is programmed to disappear," as "The highest purpose of the transsexual is to erase h/erself, to fade into the 'normal' population as soon as possible" (ibid.). Truly Trans is self-abnegating in this formulation, another liminal state to be "passed through." Stone sees as elemental to erasure "constructing plausible history" or "learning to lie effectively about one's past" (ibid.). This is reflected in all the boxes Maria needs to check to obtain hormones: "I have only ever been attracted to men, I have never fetishized women's clothes or done anything remotely kinky, I have never been sexual with the junk I was born with. Pretty much you have to prove that you're totally normal and straight and not queer at all" (pp. 41–42). Amy experiences this process in *Detransition, Baby* in at least two registers. First, medically, as she worries that her endocrinologist will declare her "not really trans" (p. 17), and thus not prescribe estrogen. Then, more anti/socially, requiring stealth as she explores her sex–gender identity at women's clothing stores, on internet story archives that "you'd have to be a certain sort of trans to ever think about looking for it in the first place. You must be this trans to ride this ride" (p. 138) and, eventually, with Reese. But Stone notes there is a Faustian bargain in rewriting history, what Mai (2018) would call fluidity with "biographical borders" in how, in what venues, and to whom one relates their past. Stone writes, "What is gained is acceptability in society. What is lost is the ability to authentically represent the complexities and ambiguities of lived experience" (ibid.). And where Mai's migrant trans sex working subjects tend to reframe their biographies in ways which materially/strategically benefit them, Stone suggests that "authentic experience is replaced by a particular kind of story, one that supports the old constructed positions" (ibid.).

But perhaps this loss of authenticity is not the only reading of constructing and navigating biographical borders. Chu and Drager (2019) read Stone with a generation of retrospect, citing the way in which Stone's work on "the rehearsed nature of trans selfnarrativizing and autobiography […] gestures to questions of authenticity" (p. 106). The response to these questions is not so much an answer as a kind of feint; while the authors seem to concur with Stone's concern regarding the collapsing of "a multiplicity of lived experiences, embodiments, and identities into one story of transness (the "wrong body" narrative), one trajectory of embodiment (medical transition), and one identity category (the passing transsexual)" (ibid.), they ultimately see Stone's manifesto as calling for the "embrace [of] transsexuality as an intertextuality, a multiplicity of genres". Trans minor literature employs a variety of strategies to thematize how authenticity can/cannot or must/must not be represented. And yet, there is something specific and important in this literature's capacity to represent community, and to de/authenticate presumptions about it.

## 4. Suffering and Authentic Representation

What is an authentic minoritarian representation, and is it different in fiction than in "reality"? What, if any, are the unique responsibilities of minor literature to minoritarian subjects and communities? Straub (2012) defines "the eternal conflict of authenticity" (p. 14) in a manner very much in line with the craft of fiction: "the externalization of the internal or the making visible and graspable what is private and on the inside" (ibid.). It is one of the reductive clichés of trans representation (the "wrong body narrative", noted above) that transition somehow visibilizes interiority. When it comes to the minor, Straub notes that "Suffering—and there tends to be suffering in narratives told from the margins of society—unmasks the individual: we expect people in pain to be at their most vulnerable,

naked and therefore authentic" (p. 17). But while Reese, Amy, and Maria may experience suffering and trauma, there is an active pushback in both novels that these experiences somehow render them more "authentically" trans. It is unsurprising that Maria tends to be wry and ironic regarding "stereotypes" around trans women; she cheekily recites: "Oh neurosis! Oh trauma! Oh, look at me, my past messed me up and I'm still working through it! Despite the impression you might get from daytime talk shows and dumb movies, there isn't anything particularly interesting there" (p. 4). She goes on to hedge, "Although, of course, Maria may be biased" (ibid.). What is the bias? That trauma is not interesting, that neurosis does not define trans representation, that a trans person need not have a messed up past to work through? Reese's friend Thalia ties history to suffering in a different way, exasperatedly mocking a "baby trans" complaining about being observed by a cis woman in a store: "That's how wounded she is, she can't take being looked at. Two eyes appraising her is trauma. I can't take it" (p. 90). The narration explains that "so devoid are these girls of their own trans history, that Thalia, having been on hormones not quite two years, has found herself forcibly placed in a maternal role" (ibid.), which is perhaps the ultimate suffering here. Does Thalia's history render her response more authentic, or (intentionally) less so?

Either way, carrying a history of being trans changes the measure of both authenticity and judgment of others' histories. Reese thinks "No matter how easily she passed as cis among the cis, passing as cis among other trans women never happened—they had trained their entire lives to see signs of transness" (p. 38). Her history heightens her awareness of passing within her own communities. *Detransition, Baby* further complicates the virtues of legibility and visibility and the extent to which these categories define "authentic" trans experience. Reese's friend Ingrid notes that the former's "incessant drama has almost nothing to do with the fact that she's trans. Her drama is just what she makes for herself as a woman" (p. 118). In this assessment, it is how Reese has decided to be a woman rather than what has been wrought upon her as trans which produces her "drama." And yet Reese wonders, with no disingenuous concern, "Without legible traumas to point to, what would pain make her?" (p. 152). It is fair to ask: would it reduce her subject position from trans woman to just woman, or, even more simply, person in pain? Does legibility of her suffering authenticate her as a trans woman, or as a subject of a novel about trans woman-ness? Reese is highly aware of her subject position justifying a right to victimhood, as she notes "among queers, trans women are still a subaltern du jour" (p. 177). Perhaps this is part of why Reese can understand or account for Ames's detransition only according to suffering: "*That is not gender*, Reese's guilt would argue, *that is pain*. All pain merits care, but not dogmatically egalitarian relativism" (p. 228). What is relative is not immediately clear, but perhaps Reese reads detransition as a kind of self-loss, what Guignon labels "releasement" (p. 8), which sees the "quest for authenticity" as often "a setup for disappointment and failure" (ibid.). This failure of authenticity is in contrast to "enowment" (p. 7), which sees authenticity as "owning oneself, of achieving self-possession." It is a one-way circuit in Reese's accounting, and by the end of the novel, Ames does not so much question his detransition as he does his self-possession and constancy. He thinks that perhaps his and Katrina's child deserves "a parent whose presence was unquestionable, because it was true" (p. 317), assuming his is questionable, and thus untrue. This thinking leads to a kind of revelation. "And finally, there, an answer: He does not want his child to know him as he is" (p. 318), which Katrina realizes, too, as she tells Ames she needs "the stability of a partner who can promise that he's more or less going to be the same person" (p. 334). By the novel's conclusion, he cannot make this promise of consistency, of fidelity to a subject position legible to her as a potential father.

The concluding section casts doubts on the prospects for minoritarian authenticity as either end or means, whether to Bettcher's liberatory theory or in line with Guignon's proclamation that "unlike happiness, authenticity is not a condition that is obviously good in itself" (p. 148). Of course, the "obvious" goodness of happiness under various definitions could be called into question for the minoritarian as well.

## 5. Authentic Doubt

Cárdenas (2017) is doubtful of not only the virtue but the actuality of minoritarian visibility. She asks and answers, "What does 'trans visibility' mean? For trans visibility to be a reality, there would have to be an essential trans identity to make visible, but there is not" (p. 170), which casts uncertainty over how to ground authenticity in any "objective" reality. She continues, "How could one make visible an identity that begins with the claim: 'I am not what I appear to be; I know this because of a feeling that I have; I am my vision of my future self.'" (ibid.). This futurity is aspirational, but also chronormatively paradoxical: I am now a vision of a future, which Cárdenas sees as contradictory with the surface, noting "Often, trans experience begins with an affective claim to futurity that rejects the truth of the visible" (ibid.). Williams (2004), picking up the thread of Lionel Trilling's thinking some thirty years earlier, suggests that "the pursuit of authenticity as a reflective ideal [...] seems to turn on a notion of honesty that links sincerity and a courageous confrontation with the truth" (p. 185), but this similarly requires a measure of truth that could be submitted to verification. The personal truths of trans minor literature often center on either dispelling stereotypes (a favorite pastime of Maria's) or playing out specific varieties of self-knowledge and observation unique to trans subject positions (the triangle of Amy–Reese–Ames reflects on matters of parentage and pregnancy, amative relationships, and gender and sexuality in sometimes contradictory ways). In the former instance, Maria plays out on her blog various "stereotypes around transsexual women" (p. 61), but not those assumed by phobic publics, those that actually "should be." Number one is that trans women are "not sex fiends, we are internet fiends" (ibid.). As Maria explains the value of the internet to trans women, she details a "whole Internet community" that dictates "stuff that's okay to talk about and stuff it's not okay to talk about, perspectives you're allowed to have and ones you're not, and its own patron saint" (p. 62). Even in the semi-anonymity of earlier internet chatrooms, blogs, and other inter-community groups, behavior is judged and circumscribed. Nonetheless, the patron saint is Julia Serano, to whom Maria applies the honorific "almost entirely unproblematic." For her part, Serano (2016) details a "form of prejudice" (p. 12) specific to trans people she dubs "cissexism" (ibid.). This prejudice holds that "transsexuals' identified genders are inferior to, or less authentic than, those of cissexuals (i.e., people who are not transsexual and who have only ever experienced their subconscious and physical sexes as being aligned)." Authenticity here is not measured as fidelity to a predetermined end, nor a journey in itself, nor, certainly, as being in some way out of step. Instead, "The justification for this denial is generally founded on the assumption that the trans person's gender is not authentic because it does not correlate with the sex they were assigned at birth" (ibid.). In this view, authenticity is predetermined by the medical industry and its relationship to the state, and even Maria's sense of Truly Trans comes up short—this medicalization acknowledges there is a "correct" and authentic way to be (and prove) trans. Serano further notes distinctions in perceiving sex–gender dissonance: "when we presume a person to be cissexual, we generally accept their overall perceived gender as natural and authentic, while disregarding any minor discrepancies in their gender appearance. However, upon discovering or suspecting that a person is transsexual, we often actively (and rather compulsively) search for evidence of their assigned sex in their personality, expressions, and physical bodies" (p. 172). This failure to pass under cisheteronormative dictates of appearance and behavior triggers questions of authenticity where none existed previously. Preciado (2008) details two categories of reaction to the scenario Serano describes. First, there are "transsexuals [who] claim to have been born 'imprisoned in the body of the opposite sex' and say that the technical mechanisms placed at their disposal by contemporary medicine are only a way of revealing their true, authentic sex" (p. 184). Second are those who "affirm their status as gender queers, or gender deviants, and refuse any summons as man or woman, declaring them to be impositions of the norm" (ibid.). Among the latter category he names Bornstein, Del La Grace Volcano, and Susan Stryker, three of the foremost thinkers on trans-subject positions in this first quarter of the 21st century. Maria understands the impact of missing a hormone

shot in terms of a reminder of her trans-ness: the absence "explains why she's been so goddam hung up on being trans. Her body is telling her, hey fucker, I am a trans body, you need to do the things that you do to take care of a trans body" (p. 51). In this sense, the "technical mechanisms" may or may not "reveal" anything authentic about Maria's sex, but serve as regularly recurring notice of her trans body.

In a different valence of medicalization, Reese's misogynistic sugar daddy notes that paying for sex with trans escorts makes him "feel bad" (p. 51). In response to her query, Stanley makes clear that paying for sex is not the issue ("What do you think this dinner is?" he plaintively replies), but that the trans escorts "all want vaginas" (ibid.). For Stanley, this is a measure of inauthenticity; these women see him only as part of funding eventual bottom surgery. He is thus satisfied at Reese's having come to terms with having a "woman's penis" (ibid.), noting that he is "interested in everything decorative on a woman" (p. 53). Reese's status as a trans woman bottom renders, for Stanley, her penis decorative rather than (sexually) functional. This, for a chauvinistic, wealthy fetishist is the du jour (literal) marker of authenticity: a woman's accepting relationship with her penis.

And yet, the obverse of "embodied" authenticity could be "performed" authenticity. Indeed, these novels give their characters plenty of opportunities to perform various sorts of authenticity. Maria is, of course, skeptical as she details her early morning face routine: "Nobody ever really gets six inches from your face and scans for stubble though, plus lots of girls have hair on their face, plus it kind of hides behind foundation a little bit, plus gender is totally 100% performative, right? Whatever! All you gotta do is perform Lady, totally embody it, and then nobody will care about anything" (p. 46). An authentic performance is one that results in no one caring about anything. Chen (2017) describes this authenticity binary: "Debates around transgender experience have been particularly inflamed around insinuations felt to be made by those averring the performative constructedness of transsexual (or, differently, transgender) identities on the one hand and the realness, the lived nature, of such identities on the other" (p. 154). Maria notes the "constructedness" of putting on her face, as well as the "realness" of no one caring—"no one" presumably being the (largely cis) publics she will encounter at the diner, her job, in bars. At another moment, though, Maria has one of her relatively few encounters with trans intimates in the form of Piranha. Maria admits, "I have a million bajillion trans things that I need to figure out, still," while "I am totally the Buddhist monk who's all convinced she's attained enlightenment!" (p. 88). What those things are remains mysterious, but at this moment it does not matter to either of them, as "It's cool that [Piranha] just lets Maria perform" (ibid.). This performance turns into a diatribe about how Maria is "stunted" despite having transitioned, the "success" of which is measured in part by "creepy old men" hitting on her in the streets. The material realities of woman-ness (creepy men) reflect back on Maria's trans reality of stunted-ness, an element of her history that persists in the novel.

She later notes the ways in which being trans implies full self-knowledge, a complete subversion of the "seeking" model of authenticity. As "more and more, the world is seeing you as a girl" (p. 99), Maria notes the ways in which trans women can embody the subject position of "het" (ibid.) women on dates. But this self-knowledge is corrupted, as, though she was acculturated to be a boy, she "[wasn't] really" as her "heart wasn't in it", and "There was an undertone of mopishness to your performance and experience of boy which isn't really there for most boys who aren't trans" (ibid.). This leads to evaluating how women treated Maria when she first began associating with them in "a non-sexual way", which is a further challenge to ever having experienced the subject position "boy" authentically.

Maria recalls Steph's observation that "Maria tends not to be very performy, and when she is performy, it's almost always for herself, not for anyone else" (p. 123). Maria attributes this characteristic to her upbringing in punk, the version of which she experienced values not caring about one's outward appearance and cultural norms thereof. But, "then you transition and realize, oh shit, there is going to have to be some intentionality in the way I present my body and my actions" (p. 124). This is an expression of minoritarian doubt regarding the prospect of authenticity as some kind of heuristic for the good life, or

liberation, or self-assurance. Maria continues, "I am going to have to break the patterns of clothing and voice and hair I've had in place all my life if I'm ever going to be read the way I want to be read" (ibid.). But does she want to be read at all, or is this admission of an inescapable social reality? Maria further admits "it would be nice to believe that you could just exist, just be some true, honest, essential self," but this belief is counterfactual, because "you only really get to have a true honest essential self if you're white, male, het, and able-bodied. Otherwise your body has all these connotations and you don't get the benefit of the doubt" (ibid.). And as noted above, even Maria's brief flirtation with white, male, het, able-bodied-ness was a farce, a poor performance.

Amy and Ames are instructive test cases for fiction's capacity to treat this embodied/performed, schismatic challenge to authenticity in shifted sex–gender identity and presentation. The "ideal of authenticity" demands a "real self" to get in touch with, but Amy's sense of her inner self is at once semi-fulfilled (as her appearance better corresponds to her sexual identification), and at the same time dissociated or unmoored. But our first introduction to this character is as Ames, in the midst of a dalliance with his boss, who, "If the genders were reversed, and some man had told his female employee to take a day off of work and come over" would "be appalled" (p. 24). Ames's passing is unquestioned; he is a man, and thus Katrina is (less) appalled at asking him to take a day off work to have sex. The sex is good, which somehow further reinforces the decision to detransition, as Ames "had transitioned to live as a woman before he had ever had really good sex, and he wasn't sure that post-detransition, he'd ever have truly good sex again" (p. 25), particularly as "Every other dalliance he'd attempted as a heterosexual man had disconnected his body and mind" (ibid.). The reader is hailed, from the outset, to wonder about Ames's detransition in terms of sexual desire: being a man apparently served him neither before or after de/transition to this point, and yet now he has found a desirable partner in a heterosexual arrangement. He, too, gets little reprieve from Maria's "thinking about being trans", as he is forced to contend with authentic sex–gender in the guise of a second opportunity to live as a cis-passing man. And yet, the grand risk of procreative sex—pregnancy—is "the one affront to his gender that he still couldn't stomach" (p. 26). His sense of authentic gender is at least initially un-responsibilized by the realities of his sexual and amative behavior.

Amy's high school experiences with women offer similar revelations. Her fumbling first attempt at sexual touch makes "obvious to the girl what Amy already knew: that there was something wrong with her masculinity" (p. 123). Amy fails an epistemic accounting of authentic masculinity: she does not confidently know how to touch a cis woman. But Delia, the first partner with whom Amy actually completes the act, signifies just how out of step any element—interior or exterior—Amy's masculinity actually is. This first experience is a different variety of dissociation: not the same sort of playacting Maria employs to (unsuccessfully) convince Steph she is getting pleasure out of their sex, but a fissure between Amy as a top during penetrative sex, and imaginative identification with Delia as a bottom. The upshot is at best unclear: "Wherever she had gone, Delia hadn't noticed. And maybe that was how you have sex" (p. 128). Amy later learns from Reese how common this type of sex is for "crypto-trans", but after the experience with Delia proceeds to sharpen "this mode of dissociative sex" (p. 130), finding a deeply asocial element in her closest physical interactions with partners. Amy's relationship with Reese feels coterminous with her identifying as a woman, but there is the dissociative, partner-frustrating sex before Reese, the heartbreak of infidelity during, and the attempts to regain intimacy with sex workers after, before eventually lapsing into the masc identity of Ames. This last iteration refuses selfnarrativizing, instead electing to embrace passing and a heteronormative coupling with Katrina. When she asks him "So you got sick of being trans?" (p. 98), Ames has obviously considered the question himself, and fractures embodied authenticity from performance again. He reveals his "damage" in a rehearsed narrative, "I got sick of *living* as trans," which implies there is something to "being" trans that is not wholly subsumed by "living as" trans. He continues, "I got to a point where I thought I didn't need to put up with the bullshit of gender in order to satisfy my sense of myself" (ibid.). The novel will refuse to

answer whether or if Ames has satisfied his sense of self, but the statement stands: "the bullshit of gender" is not the way he could accomplish this satisfaction. Put another way, separating embodiment/identity from performance: "I *am* trans, but I don't need to *do* trans" (ibid.).

Is Ames "rejecting the truth of the visible" in being without doing? Is he confronting the truth, hiding from it, willfully obscuring it? The resolve is actually slightly different from either of these: "In Ames's formulation, trans women knew what trans women were, they knew how to be, but they didn't know how to do" (p. 99). Self-knowledge, whether sought after, prerequisite, or assumed, can encompass what to be without covering what to do. One could argue that Maria decides what to do without any strong sense of what she is to be, despite the fact that she clearly has a great deal "figured out." These minor novels are perhaps antisocial in their refusal to offer answers (*Nevada* ends with James having run off with Maria's drugs, *Detransition, Baby* leaves open the questions of the baby, and any persistence of the relationship between the three principal characters). Yet they are hardly asocial in their speculations on and challenges to authenticity. Minoritarian representation in this fiction is not so much about visibility as troubling majoritarian dictates of authenticity, refusing a singular manner of being (as opposed to doing) Really, Truly Trans, and resignifying authenticity against "gender bullshit." The concept of rejecting, rather than reflecting, the primacy of the visual and visible may also be read as a refusal to pass as a foreordained minority (as opposed to minoritarian) subject position. The "damage" of authenticity is whether to acknowledge it at all, and what being ir/responsible for it does or does not signify; these remain central concerns for and defining characteristics of trans minor literature present and future.

**Funding:** This research received no external funding.

**Institutional Review Board Statement:** Not applicable.

**Informed Consent Statement:** Not applicable.

**Data Availability Statement:** Not applicable.

**Conflicts of Interest:** The author declares no conflict of interest.

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
