# Peer review of "Really, Truly Trans and the (Minor) Literary Discontents of Authenticity"

_humanities, doi:10.3390/h11060143_

Round 1
Reviewer 1 Report
On the whole I think this is a strongly-argued essay. It makes its stakes clear from the beginning, and engages effectively with the material. I have a few suggestions for strengthening it.
1. The author raises the category of "minor literature" several times. There's a tradition of literary criticism about this classification that it would be useful to outline, especially in terms of the links between minor literature on the one hand and minoritarian identity on the other.
2. While the author gestures towards answering this question, I don't think that the essay fully explores what the difference is between being "Really, Truly Trans" and passing as cis. There seem to be two different authenticating agents here: trans people who recognize others' transness even if cis people don't, and cis people in front of whom trans people successfully pass. To put it more baldly, is "Really, Truly Trans" the same as "able to pass as cis" or is it identifying with transness as a gender identity distinct from cisness?
3. The essay doesn't really explore Maria's relationship with James, which hinges in part on her ability to spot him as trans. How does that connect to the author's embrace at the end of the essay of "rejecting, rather than reflecting, the primacy of the visual and visible'? What is it about James that identifies him to Maria as trans? How does that connect to what trans is and/or does?
Author Response
- I've addressed this in the opening paragraph, inasmuch as I offer a quick sketch of the Deleuze and Guattari use of the term and what I've taken from it.
- I have added references throughout the piece the piece which I think serve to further contextualize the Truly Trans concept in light of the other ways in which I define and describe authenticity. The simplest answer, and the one I submit as a (provisional, intentionally simplistic) possibility is that "passing as cis" is almost strictly externalized, both from the individual and trans community; Truly Trans straddles a line (or perhaps modulates) between embodiment sufficient to access "gender affirming" care and recognition, and fulfilling one's own sense of what being/doing trans means. If even that distinction is messy, then so much the more successful assault on authenticity as a category for sex-gender. This is all to say, a little bit of openendedness on Really, Truly Trans is part-and-parcel to the analysis, and even "passing as cis" is a moving target depending on who is making that judgment.
- I offer a bit more attention to this side of the novel at the end of page three. I worry that much more attention would require another subsection.
Reviewer 2 Report
"Really, Truly Trans and the (Minor) Literary Discontents of Authenticity"
We should welcome this essay in an academic journal like Humanities. Often, traditional literary analysis has restricted itself to the study of aesthetic issues. In this guise, the study of transgender authenticity—what we mean when we use the label of trans to refer to ourselves or to others—has been one of those topics deemed not worth of studying in literary and "respectful" magazines. Hierarchical disciplines like literary studies do not feel comfortable with notions of fluidity and flexibility which destabilize the field of literary analysis. Not surprisingly, for most academic practitioners of literary critique, works like Nevada (2014) by Imogen Binnie and Detransition, Baby (2021) by Torrey Peters will never call their attention. They are not considered "proper" literary works. This is one of the reasons this essay should be accepted for publication in Humanities if the journal really wants to account for a field in all its manifestations instead of constraining itself to what has already been found acceptable.
Among the merits of the essay, the author has resorted to an intelligent combination of valued theoretical authors (for example, Jean-Paul Sartre and the much more thought-provoking queer philosopher, Paul Preciado) with popular culture represented by musical groups like G.L.O.S.S. (Girls Living Outside Society's Shit) or Minor Threat. It goes without saying that the essay aims at attaining a complex view of trans authenticity without looking down on any cultural manifestations. What is not conventional, like transgender labels explored in two literary works, cannot not be understood within the narrow assumptions of literary analysis. This is another reason why this essay should be published.
Even though we recommend the publication of this essay, we would like to offer a few suggestions to the author:
1. First, we implore the writer to simplify the writing style. Dropping as many adjectives and adverbs within a sentence as possible does not limit the ideas conveyed; on the contrary, it makes them more palatable to the reader. Also, not trying to insert several ideas and all its possible nuances within one sentence, makes the essay more reader friendly. I hope the author does not take this comment as an attempt at patronizing his-her-their work. On the contrary, because all of us have fallen prey to the tentation of a dense and at times obscure writing style, we feel qualified to make this suggestion. However, it is up to the author to follow or dismiss this suggestion.
2. Does the author think a brief plot summary could help those readers not familiar with Binnie's Nevada and Peters' Detransition, Baby?
3. We found some typos in the essay. A final re-reading of the paper?
After reading this essay a second time, we fully enjoyed it and appreciated the full breadth of its theoretical endeavor. My congratulations to the author.
Author Response
- I have attempted to work back through the essay and break up some longer sentences. Some sentences with longer internal quotes have been difficult to accommodate this very reasonable note.
- I've added this summary as efficiently as possible on page 2.
- I believe I have located the typographical errors, a few each in the body of the piece and the works cited.